# The Importance of Antioxidant Biomaterials in Human Health and Technological Innovation: A Review

**DOI:** 10.3390/antiox11091644

**Published:** 2022-08-24

**Authors:** Alessandra Cristina Pedro, Oscar Giordani Paniz, Isabela de Andrade Arruda Fernandes, Débora Gonçalves Bortolini, Fernanda Thaís Vieira Rubio, Charles Windson Isidoro Haminiuk, Giselle Maria Maciel, Washington Luiz Esteves Magalhães

**Affiliations:** 1Embrapa Florestas, Colombo 83411-000, Paraná, Brazil; 2Programa de Pós-Graduação em Engenharia de Alimentos (PPGEAL), Universidade Federal do Paraná (UFPR), Curitiba 81531-980, Paraná, Brazil; 3Departamento de Engenharia Química, Universidade de São Paulo, Escola Politécnica, Sao Paulo 05508-080, Sao Paulo, Brazil; 4Laboratório de Biotecnologia, Universidade Tecnológica Federal do Paraná (UTFPR), Curitiba 81280-340, Paraná, Brazil; 5Programa de Pós-Graduação em Engenharia e Ciência dos Materiais—PIPE, Universidade Federal do Paraná, Curitiba 81531-990, Paraná, Brazil

**Keywords:** antioxidant biomaterials, biological properties, technological applications

## Abstract

Biomaterials come from natural sources such as animals, plants, fungi, algae, and bacteria, composed mainly of protein, lipid, and carbohydrate molecules. The great diversity of biomaterials makes these compounds promising for developing new products for technological applications. In this sense, antioxidant biomaterials have been developed to exert biological and active functions in the human body and industrial formulations. Furthermore, antioxidant biomaterials come from natural sources, whose components can inhibit reactive oxygen species (ROS). Thus, these materials incorporated with antioxidants, mainly from plant sources, have important effects, such as anti-inflammatory, wound healing, antitumor, and anti-aging, in addition to increasing the shelf-life of products. Aiming at the importance of antioxidant biomaterials in different technological segments as biodegradable, economic, and promising sources, this review presents the main available biomaterials, antioxidant sources, and assigned biological activities. In addition, potential applications in the biomedical and industrial fields are described with a focus on innovative publications found in the literature in the last five years.

## 1. Introduction

There is growing interest in the biomaterial field, which combines expertise from many disciplines, including materials science, physics, engineering, chemistry, and medicine [1]. A biomaterial is a synthetic or natural material intended to interact with biological systems and can direct the course of any therapeutic or diagnostic procedure. Its concept and application have been expanded from medical instruments to medical products, including implantable and interventional devices, in vitro diagnostic agents, and drugs and biological products [2,3].

If a biomaterial is implanted, it typically triggers some degree of inflammatory response and accompanying healing. If it turns into an excessive inflammation, reactive oxygen species (ROS) are ordinarily produced to regulate the healing response. Therefore, the biomaterial’s function may be negatively affected both short and long term [4]. ROS are products of oxygen metabolism identified as molecules with unstable bonds or species with one or more unpaired electrons on the external orbital. Besides the endogenous origin, the generation of these species can also occur via exogenous causes and provides instability, causing exceptional reactive behaviors. When ROS production increases, the balance between their amount and the antioxidant defense is altered, and oxidative stress occurs [5]. This imbalance has implications for many disease states, including cardiovascular damage, inflammation, cancers, and neurodegenerative diseases [6].

Antioxidants are well-known compounds that can play a significant role against these disorders, eliminating harmful ROS excess, and inhibiting or delaying molecular oxidation. Therefore, to overcome the problems arising from oxidative stress, the development of biomaterials with antioxidant properties occupies an increasing segment in the current panorama. Furthermore, engineering antioxidant properties into a material may be an important goal to improve biocompatibility and propose a solution against oxidative degradation. In this sense, there is a growing effort toward searching for antioxidant molecules that are non-toxic and can be used as antioxidants and preservatives in food, cosmetics, and medicines, as well as in different therapeutic applications [5].

Antioxidant biomaterials have been used in drug delivery systems, soft tissue regeneration, chronic wound healing, and cosmetical applications. Besides using antioxidant materials in biomedical applications, they can also be incorporated into widely used polymers for stabilization and functionalization. Antioxidants can increase the stability of conventional polymer by inhibiting photo and thermooxidative degradations without undesirable effects associated with releasing toxic additives. Likewise, studies regarding functionalization are also important, especially for active food packaging development, aiming to delay or prevent oxidative deterioration of packaged food [6].

This review presents an update of currently available biomaterials endowed with antioxidants and provides an overview of their main biological properties and applications in various sectors.

## 2. Biomaterials

Biomaterials are derived from synthetic or natural renewable sources. They must present characteristics such as immunogenicity, availability, low production cost, and resistance to sterilization processes to be considered biodegradable, economical, and promising alternatives for applications in different industrial areas [7]. The conscious production of biomaterials presents a viable and sustainable alternative for replacing packaging [8] and textile materials [9], as well as applications in the areas of biomedicine and dentistry [10,11]. Some biomaterials show biological properties derived from raw material sources. These biomaterials have attracted investigators because of their antioxidant properties. Therefore, this review is focused on antioxidant biomaterials from natural sources for health and food applications.

Natural biomaterials can come from living organisms such as animals, plants, fungi, algae, and bacteria (Figure 1). Thus, there is a diversity of biomaterials spread throughout nature, which can be formed by chains of already known molecules such as proteins, carbohydrates, and lipids.

Proteins are produced especially by members of the animal kingdom; collagen, keratin, and gelatin, in particular, are used for pharmaceutical and food applications. Collagen and keratin are long filamentous fibrillar proteins that make up the extracellular matrix of animal cells, known as scleroproteins. The versatility of these biomaterials is reflected in diverse areas of application in biomedicine, such as wound healing, tissue engineering, surface coating of medical devices, and skin supplementation. In this sense, the biocompatibility of these proteins contributes to the successful application of these biomaterials [11]. Furthermore, gelatin, a protein characterized as partially hydrolyzed collagen, has wide applicability in food production, to modify texture, retain water, foaming, emulsifying, and stabilizing colloidal systems [12]. In addition, gelatin can be used as an encapsulating agent, as a delivery system for drugs and bioactive compounds, and in the production of biological tissues, due to its biocompatibility, non-toxicity, and biodegradability [13].

Fatty acids and waxes are produced by insects such as bees. These components can be used to develop solid lipid nanoparticles used as drug delivery systems for different medications [14].

The biomaterials derived from carbohydrates are chiefly presented as polysaccharides, which plants, bacteria, and algae can originate. Cellulose, consisting of a chain of glucose joined by β-(1→4) bonds, can be produced by plants and bacteria. The use of cellulose as a biomaterial involves the production of antimicrobial, biodegradable, and antioxidant packaging, emulsifier, texture modifier, drug and polyphenol delivery systems, skin care products, and heart valves and blood tubes [15]. Starch, a chain of glucose residues linked together by α-(1→4) glycosidic bonds (amylose) and ramified by α-(1→6) glycosidic bonds (amylopectin), is formed as an energy reserve in plants. Starches have a gelling capacity and, therefore, are widely used in the food industry as texture modifying agents. In the case of biomaterials, starches can be used in the production of biopolymers such as ethylene and polyethylene, contributing to the reduction of consumption of materials derived from petroleum [16]. Chitin and chitosan, widely used in tissue engineering due to their biocompatibility, can be produced by some fungi and crustaceans. Chitin is a semicrystalline homopolymer of β-(1→4)-linked N-acetyl-D-glucosamine. Chitosan is obtained by boiling chitin in potassium hydroxide, making it soluble in organic acids [17]. Gelling polysaccharides such as alginate, agarose, and carrageenan are mainly extracted from algae. They have been used industrially as thickeners, emulsifiers, gelling agents, and film-forming agents, offering a unique texture for food products, with significant application also in biotechnological or biomedical areas [18,19,20].

All of these biomaterials can be produced or used as inspiration for the production of similar artificial materials in order to develop new products for industrial and medical applications. However, natural sources can improve biological properties, such as the antioxidant activity of such biomaterials.

Among the biomaterials that have been developed, antioxidant biomaterials are promising in this area [21,22]. These biomaterials can inhibit excess superoxide, hydroxyl, hydroperoxyl, and hydrogen peroxide radicals, also known as reactive oxygen species (ROS), which characterizes their antioxidant capacity. ROS are essential for the regulation of metabolism. However, their excess, caused by several extrinsic factors, can affect their healthy functioning. Thus, antioxidant biomaterials may have several positive health effects, for example, the treatment of inflammatory diseases [23], metabolic diseases [24], and wound healing [25]. The topic of antioxidant biomaterials and their biological effects will be explored in detail in the following sessions.

According to the Web of Science database [21], there are 1460 registered publications searched through the “antioxidant biomaterials” keyword, of which 168 are literature re-views. These publications are divided by field, highlighting Materials Science Biomaterials, Engineering Biomedical, Polymer Science, Materials Science Multi-disciplinary, and Biochemistry Molecular Biology (Figure 2B). Furthermore, there is a prospect of growth in the number of publications per year, as shown in Figure 2A. Thus, antioxidant biomaterial technology is a promising alternative for improving materials science technology.

### 2.1. Antioxidant Biomaterials

The accumulation of ROS is responsible for the lipid peroxidation of the hydroxyl radical, hydrogen and superoxide anion. ROS are associated with alterations and damage to cellular constituents (membrane lipids, nucleic acids and proteins), which signal several complications for human health (inflammation, proliferation and differentiation of stem cells, chronic wounds, bone defects, Parkinson’s and Alzheimer’s diseases, among others) and negative implications for food (reduction of shelf life, change in taste, aroma and odor) [26,27]. In this context, natural antioxidant biomaterials become promising to inhibit or delay molecular oxidative stress through the release of compounds with antioxidant activity, as shown in Figure 3 [26].

Thus, the functionalization of these biomaterials becomes a promising strategy for application in different technological segments, such as tissue regeneration engineering, active packaging for food, and hydrogels for the treatment of chronic diseases and inflammations (applications described in detail in Section 4 of this review article). In addition, new smart biomaterials are being developed to programmably modulate oxidative stress [5].

#### 2.1.1. Carriers and Antioxidants

##### Marine Organisms

The use of marine organisms to develop antioxidant biomaterials has also been well explored to remove ROS responsible for molecular oxidation. For example, in the study by Wang et al. [27], hydrogels were formed based on recombinant thrombospondin antioxidant protein (rich in cysteine residues), which is extracted from adhesive components of sea anemones. These hydrogels were able to prevent cellular oxidative damage, decrease lipid oxidation in the skin of mice, and stimulate endogenous antioxidant systems. In addition, they showed biodegradability, biocompatibility, and moisture resistance.

Carrageenan films (sulfated polysaccharides from red seaweed), ulvan hydrogels (sulfated polysaccharides from green seaweed), and alginate hydrogels (obtained from brown seaweed) also show numerous bioactive antioxidant properties and promising viscoelastic properties for use in medical applications, food or even as anti-polluting agents [27,28,29]. Like alginate, chitosan polymers (polysaccharides extracted from the exoskeletons of crustaceans) are the matrices most used as biomaterials associated with antioxidants. The study by Kaczmarek-Szczepańska [30] developed antioxidant 3D scaffolds by cross-linking chitosan and collagen polymers in glyoxal solution. After loading the matrix with melatonin, this biomaterial presented maintenance of mitochondrial homeostasis and a cleaning of ROS under stress conditions, offering, therefore, protection of the cells.

The low molecular weight chitosan production gives the biomaterial greater solubility at different pHs, swelling capacity (0.2–2.0 mg/mg) required for an effective delivery system, and an enhancement of radical scavenging activity (58–75%). This type of chitosan can be obtained mainly by depolymerization from the solution plasma process [31]. Despite the versatility, rapid cross-linking, and water solubility characteristics of chitosan itself, its derivatives are also effectively employed as antioxidant biopolymers [32,33,34,35].

In this context, Ali et al. [32] developed a biopolymer derived from chitosan imine through chemical modification (Schiff base). As a result, the antioxidant activity showed the potential to reduce intracellular ROS and stop bacterial growth, as well as non-cytotoxic and biocompatible characteristics. Jafari et al. [33] used the oxidative degradation of chitosan by means of microwave-assisted irradiation to produce the bioagent chitooligosaccharide, whose developed films showed antioxidant activity, improved biological activities, cytocompatibility and a positive effect on fibroblast migration. Zhou et al. [34] also produced films from tyrosine residues in silk fibroin (*Bombyx mori*) enzymatically oxidized using laccase; later, the Schiff base reactions and Michael addition promoted the coupling of chitooligosaccharide. This functionalization was responsible for the antioxidant characteristic of the biomaterial, which was able to improve 2,2′-azino-bis(3-ethylbenzothiazoline-6-sulfonic acid) (ABTS) free radical scavenging.

Tan et al. [35] incorporated α-lipoic acid into chitosan acetate films to improve its antioxidant effect, which had a reducing effect on superoxide, hydroxyl anions and DPPH radicals. This characteristic is associated with the action of α-lipoic acid, which promotes the amortization of oxygen free radicals, chelating metals, recovery of oxidized antioxidants, as well as aiding the functionality of enzymes with antioxidant properties.

The association of different polysaccharides has become an alternative strategy to improve the synergistic properties of composite films. Don et al. [36], for example, developed chitosan films with ulvan, whose biomaterial showed antioxidant activity and bleaching properties and controlled ulvan release of 40–65% for 12 h. Tan et al. [37] developed composite films of chitosan ascorbate and methylcellulose, which exhibited barrier properties against visible ultraviolet light, excellent mechanical properties and increased free radical scavenging.

#### 2.1.2. Antioxidants

##### Microorganisms and/or Plant Extracts

Copolymers produced by microorganisms and functionalized with antioxidants are also promising as biomaterials. Generally, these biopolymers, based on poly(3-hydroxybutyrate-co-3-hydroxyvalerate)-P(3HB-co-3HV), poly(3-hydroxybutyrate-co-4-hydroxybutyrate)-P(3HB-co-4HB) and poly(3-hydroxybutyrate-co-3-hydroxyhexanoate)-P(3HB-co-HHx), can be obtained from *Escherichia coli* [38].

In the study by Bhatia et al. [38], the copolymer poly(3-hydroxybutyrate-co-3-hydroxyvalerated)-P(3HB-co-3HV) was functionalized with ascorbic acid, whose biomaterial presented antioxidant effects, biodegradability, better mechanical properties, thermal stability (thermal degradation at 295 °C) and potential as a low-cost alternative to petroleum-based polyesters. Similarly, Roy and Rhim [39] developed functional poly(lactic acid) films incorporated with curcumin, which also showed antioxidant effects, biodegradability and better mechanical properties. In addition, the films exhibited barrier properties against ultraviolet (UV) rays and water vapor. This barrier property against UV rays is associated with the presence of chromophores, organic functional groups that establish the conversion of visible and UV spectra into small differentiated molecules, in the structure of phenolic compounds responsible for the antioxidant activity of plant extracts such as curcumin.

Polyphenols from different plant sources have been widely used in biomedicine for the prevention and treatment of cardiovascular, neurodegenerative, bone and cancer diseases, due to their antioxidant properties and targeting of molecular pathways involved in the generation of ROS [5]. Phenolic compounds such as ferulic acid, catechin, and cyanidin-3 from extracts of leaves and flowers (*Camellia sinensis*, *Yerba mate* and *Hibiscus*) and fruits (acerola, melon, kiwi, tomato, apple and pomegranate) have shown an antioxidant capacity, a determinant for their use for photoprotection against UV rays [40]. Mainly, when incorporated into bacterial cellulose membranes, whose biopolymer is produced especially by the strain *Gluconacetobacter xilynus*, their production can be optimized using several parameters such as choice of fermentation method, production conditions (substrate, co-substrate, pH, temperature, dissolved oxygen and inoculum rate) and statistical methods of experimental design [16].

Studies have shown that polyphenols can contribute to the mechanical, biological and degradation properties of polymers intended for packaging and bone tissue regeneration. The study carried out by Zhao et al. [41] showed that polyphenols extracted from *Fructus chebulae* are useful for biomedical and pharmacological applications, due to the presence of pyrogallol and hydrolysable tannins, which have phenol carboxylic acids of high nutritional value and high antioxidant activity. These polyphenols were efficiently incorporated into gelatin hydrogels as crosslinking agents, providing improved physical properties to the hydrogel. Multifunctional and regenerative hydrogels developed with the addition of green tea extract (*Camelia sinensis*) showed mechanical strength, biocompatibility and antioxidant activity via ROS assay. In addition, they helped in the healing of diabetic wounds, facilitating the pro-angiogenic properties [42].

Studies show that flavonoid-based polymeric materials may have better antioxidant and antimicrobial properties compared to monomeric polyphenols [43]. The flavonoid epigallocatechin gallate (EGCG) is available in high concentrations in red to violet colored fruits. Polycaprolactone films coated with EGCG increased cell adhesion and were effective for dermal tissue regeneration [44]. Another flavonoid with important applications in biomaterials is naringenin, a natural and antimicrobial additive obtained mainly from citrus fruits [43]. The oxidation resistance, thermal stability, and free radical and Cu^2+^ ion scavenging ability of naringenin cross-linked to glycerol diglycide ether makes this biomaterial promising for application in polymeric packaging with active properties.

The Kerifran exopolymer has also been noted for its antioxidant and probiotic properties. This biopolymer can be produced from kefir grains, from restricted cultures of *Lactobacillus kefiranofaciens*, or from mixed cultures of *Lactobacillus kefiranofaciens* with *Saccharomyces cerevisiae*, using anaerobic or aerobic conditions and/or ultrasound extraction [45].

Natural polysaccharides can be excellent biocompatible biomaterials for applications in the food, biofuel, water purification and pharmaceutical industries. Inulin, for example, is extracted mainly from roots and tubers (onions, garlic, artichokes and yacon) and has antioxidant activity. However, its derivatives (amino-pyridine, o-(aminoethyl) inulin, N-(aminoethyl) inulin and diphenyl phosphate) have this potentiated bioactivity, as is the case for the polysaccharide diphenyl phosphate, which, compared to inulin, has high antioxidant activity against hydroxyl radicals (98.2%), DPPH radicals (1.6 mg/mL) and superoxide anion radicals (95.4%) [46].

Arbutin, a glycosylated hydroquinone belonging to the Ericaceae family, is often found in mitrile, cranberry, uva ursi, strawberry, as well as cereals such as wheat. This antioxidant has biological properties related to skin whitening and antimicrobial effects, being potentially applied in the biomedical area. Arbutin-based polymeric coatings showed good interaction with osteoblast precursor cells and suggested high biocompatibility [47].

##### Nanoparticles

Antioxidant biomaterials can also be used as a nanoparticle delivery systems. Sebastiammal et al. [48] studied the use of curcumin in alginate to encapsulate hydroxyapatite nanoparticles (biological molecule and mineral component found in bones), which showed excellent free radical scavenging efficiency.

Vinay et al. [49] developed an antioxidant biomaterial under the hydrothermal synthesis method from *Elaeocarpus ganitrus* seed extract with gold nanoparticles, whose covering and reducing agent constituted by the extract rich in tannins, quercetin, flavonoids, ellagic acid and gallic acid, reduces chloroauric acid (HAuCl_4_) into gold nanoparticles.

In the same way, copper nanoparticles can also be synthesized for the development of bioactive and multifunctional hydrogels. The study by Gong et al. [50], for example, aimed to produce a biomaterial with F127-copper nanoparticles using the in situ coordination method. The formation of the antioxidant hydrogel occurs through the interaction of sodium phytate (plant-based metal ion complexing agent) with the Cu^2+^ of the F127 aqueous solution. Thus, this hydrogel has the ability to efficiently reduce 2,2-diphenyl-1-picrylhydrazyl (DPPH) free radicals and reduce oxidative stress caused by lipid peroxidation.

##### Other Antioxidant Sources

The incorporation of commercial antioxidants into biomaterials also presents significant potential for drug delivery systems and antioxidant solutions. In the study by Fan et al. [51], for example, the authors incorporated Edaravone^®^ polyacrylic ester nanoparticles in alginate to be employed as a topical hydrogel of the commercial antioxidant. This biomaterial was able to present an increase in solubility (5 mg/mL), loading efficiency and prolonged release of the antioxidant, as well as an inhibition of lipid peroxidation.

Inorganic-organic hybrid biomaterials based on rare earths show promising antioxidant activity associated with their composition of seventeen metals consisting of scandium and yttrium and fifteen elements of lanthanum. This biomaterial, when incorporated into hydrogels, nanoparticles, nanofibers or porous scaffolds from materials such as alginate, gelatin, chitosan, poly-(ε-caprolactone), polyacrylonitrile, poly(vinyl alcohol), polylactic acid and polyhydroxybutyrate, can be used for biosensor, drug delivery, photodynamic therapy and tumor theranostics applications [52].

Liquid styrax from Anatolian sweet gum (*Liquidambar orientalis* Mill.) has been used for the development of cryogel scaffolds [53]. This biomaterial loaded with liquid styrax has important antioxidant and antimicrobial properties, due to its ferrous ion chelating activity, with potential application in tissue engineering.

Biomaterials such as cryogels obtained from hyaluronic acid modified by adipic acid dihydrazide have also been highlighted as multiphase biological dressings due to their cytocompatibility, regulation of reactive oxygen species (antioxidant activity) and stable mechanical strength, since their macroporous structure promotes a high swelling property required for fluid delivery and/or removal systems [54].

Furthermore, they may depend on the cross-linking between the adsorbent matrix and the adsorbate explained by electrostatic interactions [40], resulting, therefore, in a search for antioxidant biomaterials that present high rates of release of their components in a continuous and prolonged way [45,55], ensuring, then, that antioxidant biomaterials can play a key role in a wide range of applications, including nanodrug delivery systems, cell preservation (hypothermic and cryopreservation), food preservation (pathogen control agents, packaging and biosensors), edible packaging or food stabilizers, water treatment, orthopedic and dental coverings, wound dressing, and cosmetics with a protective and/or anti-aging effect [55,56].

## 3. Biological Properties and Release of Antioxidants from Biomaterials

Biomaterials can be used as carriers of antioxidants or active principles for cells and tissues with oxidative damage. For the antioxidant therapy to be effective, the delivered antioxidant must scavenge the correct oxidative species, be delivered directly to the target tissue, reach effective levels at the intended site, and remain functional as long as pathologically relevant oxidative stress is present [4].

It is important to relate the release of antioxidants with the biomaterial’s biodegradability and its type of in vivo application. Some biomaterials require long-lasting duration in tissues or permanent integration into them; thus, the controlled release of antioxidants can be achieved more or less slowly in balance with the scaffold’s physical features and biodegradation [57]. Many loaded antioxidant components are released too quickly, are released in incomplete form, or are unstable during the release process [27].

Encapsulated or entrapped antioxidant compounds often exhibit poor long-term stability, and most of time, there is a need for sustained, long-term and targeted therapies. Encapsulation strategies usually permit only limited depots of antioxidants and function primarily through passive release or diffusion of antioxidant compounds, limiting their potential for prolonged and controlled antioxidant therapy. Therefore, smart systems seem an interesting approach, where particles only release their antioxidant payload when triggered by supra-threshold levels of reactive oxygen species, offering the advantages of protecting the antioxidant from inactivation, providing continuous antioxidant protection and relatively high mass of antioxidant payload [4].

In general, the release of bioactive compounds depends on solubilization and diffusion in the biomaterial and solubility of the compound in solution. The biomaterial matrix with different pore structures and water solubility plays a crucial role in the release of bioactive compounds [39].

Studies demonstrate different delivery systems of antioxidants incorporated in biomaterials. Gelatin microspheres loaded with curcumin nanoparticles were developed by Liu et al. [58]. Curcumin release was greater in the presence of matrix metalloproteinases, which are generally overexpressed in non-healing diabetic wound sites (Figure 4). Thus, the system was responsive to these metalloproteinases, and curcumin was released specifically at the sites to enhance bioactive effects. Another curcumin-related study by Roy and Rhim [39] evaluated the release of curcumin from poly (lactic acid)-based functional films. The authors reported a slow curcumin release from the films that started after a few hours of immersion in water. Furthermore, the release was low, probably due to the low water solubility of curcumin.

Wu et al. [59] modified polyvinyl alcohol using citric acid, aiming for the development of a biomaterial model for therapy of degenerative and inflammatory diseases. Citric acid was released as a co-oligomer with carbon chain fracture and autopolymerization. According to the authors, both can degrade into stable molecular monomers (citrate) over time, which can provide relatively high antioxidant content and persistent local antioxidant release. In the work carried out by Fan et al. [51], edaravone-loaded nanoparticles and nanocomposite hydrogels based on alginate were prepared. Both formulations displayed sustained release of edaravone over a period of 9 h, and edaravone released from nanocomposite hydrogel presented a longer sustained release pattern, which was related to the complex structure of the matrix. The authors explained that, within the nanocomposite hydrogel system, edaravone should cross two physical barriers, the inner Eudragit nanoparticles and the outer alginate hydrogel.

Thus, the release profiles of biomolecules from polymeric systems can be controlled by diffusion, polymer degradation, dose of antioxidant loaded in the system and biomaterial composition (Figure 4).

Considering the delivery systems of antioxidant molecules described above, antioxidant biomaterials may contributes to important biological properties in the human body, such as anti-inflammatory, healing, anti-aging, antitumor, immunomodulatory, and antimicrobial effects, in addition to tissue regeneration and reconstruction (Figure 5).

However, the systemic delivery of antioxidants by biomaterials can be influenced and hampered by some factors related to malabsorption, loss of antioxidant activity and efficacy [4]. As a result, new biopolymers incorporated with antioxidants have been developed to maintain the bioaccessibility and bioavailability of antioxidant compounds, exerting different biological functions in the human body (Table 1).

The study carried out by Wu et al. [59] showed the effects of antioxidant biomaterials based on polyvinyl alcohol modified with citric acid (PVA-C) on degenerative diseases induced by reactive oxygen species. The results showed that the citric acid-modified polymer allowed the regulation of redox signaling of mesenchymal stem cells (BMSCs) through peroxisome proliferator-activated receptor nuclear receptor γ (PPARγ) and superoxide dismutase (SOD). Furthermore, in vivo assays showed that the PVA-C biomaterial inhibited oxidative stress and inflammatory reactions induced by lipopolysaccharides (LPS).

Oxygen free radicals are also produced by physical, thermal, or physiological damage to the skin. These reactive species increase cell wall permeability, causing damage to vascular endothelial cells and cell necrosis, which makes healing difficult. In this context, several studies have been carried out on the application of antioxidant biomaterials for wound healing. Xu et al. [65] demonstrated that nanogels, produced from polyurethane copolymers of polyethylene glycol (PEG), polypropylene glycol (PPG) and polydimethylsiloxane (PDMS) and incorporated from lignin extracted from coconut shell, can be incorporated as antioxidant materials in dressings and accelerate the healing process of skin wounds. In vivo studies showed that lignin-embedded nanogels increased the expression of the cell proliferation marker Ki67, causing the acceleration of wound healing in mice. Furthermore, these biomaterials reduced the active oxygen level and protected human hepatocyte cells (LO2) against apoptosis. Other studies show that hydrogels loaded with natural (jujube extract [63], curcumin [58], Edaravone^®^ [51] and ulvan—green macroalga [29,36]) or modified (modified dopamine [66]) antioxidants, used alone or combined, can be efficiently applied in an on-demand delivery system in wound healing in diabetics. Chen et al. [60] also developed hydrogels composed of alginate-chitosan and tetracycline gelatin microspheres. Different concentrations were tested, and hydrogels at 30 mg/mL showed better mechanical and stable properties for wound healing. In addition, the developed biomaterials showed significant inhibition of *Escherichia coli* and *Staphylococcus aureus* bacteria.

Incorporated biomolecules of antioxidant compounds, in addition to healing skin lesions of varying severity, can slow down the natural process of human aging through skin vascularization and migration of keratinocytes and fibroblasts. A biocosmetic produced from peptide-conjugated caffeic acid reduced damage to the skin cell membrane induced by oxidative stress [61]. Antioxidant biomaterials based on a specific invertebrate protein were recently discovered as a potential tissue regenerator. Wang et al. [29] developed hydrogels with the type 1 cysteine-rich thrombospondin-1 protein (TSRL) with stronger 1,1-diphenyl-2-picrylhydrazyl (DPPH) radical scavenging rates than glutathione and ascorbic acid. The TSRL hydrogels were applied to the skin of mice and demonstrated a decrease in epidermal hyperplasia and in the degradation of collagen and elastic fibers caused by the ultraviolet B (UVB) radiation.

Studies have shown that mitochondrial metabolism also influences the increase in reactive oxygen species (ROS), which contribute to the induction and progression of tumor cells. Local treatment methods (localized radiotherapy, chemotherapy or phototherapy) assisted by antioxidant biomaterials can stimulate the immune system, through immunological memory, and cause the death of metastatic cancer cells. Thus, these immunomodulatory therapies with biomaterials could prove to be alternative strategies to improve therapeutic responses against cancerous tumors [69,70].

Sebastiammal et al. [48] showed that nanoparticles (NPs) of hydroxyapatite (HAp), a biological molecule and the main mineral component found in bones, incorporated with curcumin have an anticarcinogenic effect against the HeLa cell line (cervical cancer), being effective in biomedical applications. A minimum concentration of 100 µg/mL of HAp NPs was sufficient to cause 30% of cell death. As a biomaterial, HAp NPs have a greater ability to adhere to cancer cells than to normal cells, based on electrostatic interactions between negatively charged sites on cell membranes and positive binding sites on the HAp surface. Other biological activities, such as effects against *Shigella flexneri*, *Escherichia coli*, *Pseudomonas aeruginosa*, *Klebsiella pneumonia* and *Staphylococcus aureus* bacteria, were observed for HAp NPs nanoparticles. Similarly, Vinay et al. [49] developed an antioxidant, antimicrobial and anticancer biomaterial from *Elaeocarpus ganitrus* seed extract with gold nanoparticles. The important biological activities of this material make it highly viable for applications in different fields such as water treatment, food preservation, dressings, nanomedicines, cosmetics, biocides and disinfecting agents.

Ivanova and Yaneva [22] showed that different studies with chitosan-based biopolymers exhibited an anticancer effect. The mechanisms of anticarcinogenic action were associated with activities of redox modulation (human defense system), depolarization of the mitochondrial membrane and activation of caspases responsible for inducing apoptosis. In addition, chitosan-based biomaterials were associated with immunostimulatory activities related to the alteration of calcium homeostasis in tumor cells. Other studies show that biomaterials developed with chitosan have an antimicrobial effect, mainly against *Staphylococcus aureus* and *Escherichia coli*, caused by increased positive charges and electrostatic interactions with cellular components [32,33,34,35,36]. Fakhri et al. [67] cite chitosan biomaterials in the production of biodental materials, oral drug delivery systems and bone tissue engineering for dentin-pulp regeneration processes and treatment of periodontitis. The versatile biological properties, biodegradability, biocompatibility and non-toxic nature make this biopolymer innovative and physiologically active for different applications.

Antioxidant biomaterials have also shown positive effects in the reconstruction and regeneration of cellular tissues. Furthermore, these materials can prevent bacterial infections associated with medical devices introduced into the human body. Baldwin et al. [68] studied the in vivo biocompatibility of injectable tannic acid and collagen spheres for post-lumpectomy breast tissue reconstruction. After 12 weeks, the tannic acid and collagen-based biomaterials showed adipose tissue regeneration and collagen redistribution in the breast, indicating good biocompatibility and bioactivity. In addition, no infections, tissue necrosis or chronic inflammation disseminated by the implantation of these biomaterials were observed. Liu et al. [64] developed biomedical catheters with tannic acid and benzalkonium chloride (hydrophobic agent) coatings in order to prevent bacterial infections. The developed biomaterial showed excellent bactericidal activity against *Staphylococcus aureus* and *Escherichia coli*. Thus, hydrophobic tannic acid easily formed a stable, colorless coating on the luminal and external surfaces of catheters with antibacterial and biocompatible properties.

The biological properties described in the studies discussed above make antioxidant biomaterials important additives for the development of products in the biomedical, chemical, cosmetic, pharmaceutical and food areas, as described in detail in Section 4 of this review.

## 4. Industrial and Technological Applications

Antioxidant biomaterials, as mentioned in the topics above, are studied extensively due to their potential applications in various industrial and technological areas with regard to materials susceptible to oxidative degradation that require antioxidant protection. Thus, providing a recent compilation of studies related to antioxidant biomaterials, applications in the food, biomedical and pharmaceutical fields are discussed in this section [71].

### 4.1. Active Packaging

Packaging materials must present, in addition to basic properties such as tensile and thermal resistance, characteristics that contribute to stability and shelf life, such as antioxidant and antimicrobial agents [72]. Food quality and safety are important factors for the development of active packaging based on biomaterials. Active packaging increases shelf life through antioxidants, which contribute to food protection and quality based on international traceability standards [73].

The oxidative process of lipids can result in changes in odor, color and texture in food products [74,75]. Oxidation negatively alters nutritional and sensory quality and food safety. As a result, new technologies have been developed through the incorporation of additives with antioxidant properties to food packaging.

Studies show that packages containing lignin can extend the shelf life of foods by inhibiting the action of free radicals and oxidation damage. Fontes et al. [75] developed films of poly(lactic acid) (PLA) and lignin nanofibers extracted from rice husk by the Organosolv method. The biomaterials showed good conditions of thermal stability and hydrophobicity and high antioxidant activity. Active packaging and aerogels based on corn starch and pine nut husk extract, produced by electrospinning and lyophilization, were considered effective for food packaging. In addition, the starch-encapsulated extract showed bioaccessibility when subjected to in vitro gastrointestinal digestion, being considered a promising functional food [76,77,78].

A versatile and widely used biomaterial in active packaging is chitosan. It is a renewable biopolymer, being the second most abundant polysaccharide in nature after cellulose. It has good film forming ability, biodegradability, non-toxicity, and excellent antioxidant and antimicrobial properties. However, some factors such as mechanical strength and barrier properties need to be improved [79,80,81,82].

In order to improve the performance of chitosan films, studies have produced blends from the incorporation of other compounds. Chitosan films with polyvinyl alcohol, guar gum and moringa extract (bioactive rich in compounds with antioxidant, antitumor, anti-inflammatory, antibiotic and antimicrobial properties) showed superior mechanical performance compared to films without the extract. This behavior was attributed to the formation of intermolecular hydrogen bonds between functional groups of the polymers and phenolic and carboxylic groups of the extract [82].

Another relevant point concerns the antioxidant activity of pure chitosan, as it does not reach the minimum standards for use in active packaging, and the addition of other compounds is also necessary. Nanocellulose films oxidized with TEMPO and incorporated with chitosan showed a high antioxidant capacity when compared to control nanocellulose (without chitosan incorporation) [82]. Graphene oxide nanoparticles were incorporated into chitosan films, promoting an increase of up to 82% in antioxidant capacity, as well as a significant improvement in mechanical and electrical properties. The developed films showed potential application in food packaging, body sensors and electro-responsive devices [78]. Silica and β-acids from hops were incorporated into the chitosan matrix, aiming to protect soybean oil against lipid oxidation and the presence of pathogens [80]. In addition, baicalein was also incorporated into the chitosan matrix, causing improvement in physical and antioxidant properties, in addition to inhibiting the oxidation of soybean oil [81]. The extract of broken grains of Thai rice (*Oryza sativa* L.), rich in polyphenols and anthocyanins, was incorporated into the chitosan matrix for the development of packaging. The developed films proved to be a good visual tool to indicate the freshness of seafood (pH indicator), ensuring food quality and safety [83]. Shen et al. [84] incorporated curcumin into chitosan films. The edible packaging developed showed an effect against the lipid oxidation of pork meat.

Thus, different studies in the literature present the importance of antioxidant biomaterials for applications in the food industry, mainly as active packaging (Table 2).

### 4.2. Biomedical Applications

Wounds are breaks or deformations in the skin caused by physical, thermal, medical or physiological damage [94]. ROS interfere with wound healing, increasing cell wall permeability and causing damage to vascular endothelial cells, necrosis and tissue dissolution [94]. In addition, ROS, associated with inflammatory principles, can cause effects of premature aging, neural disorders, cardiovascular diseases and cancerous tumors [95]. Studies show that antioxidant biomaterials can inhibit the damage caused by ROS [96], as biomaterials can be used as carriers of antioxidants for tissues that are under oxidative stress, as well as drugs and cells [97,98].

A compound that has high potential for use as an antioxidant agent in biomaterials is lignin, due to the large amount of phenolic compounds and phenylpropane units with -CH_2_ side units [75,94]. Due to the antioxidant properties mainly of low molecular weight lignins, several studies have explored their potential application in biomedicine, such as dressings and grafts. A nanogel of poly(dimethyl siloxane) (PDMS), polyethylene glycol (PEG), polypropylene glycol and lignin-incorporated hexamethylene diisocyanate, extracted from coconut husks, showed high antioxidant activity, compatible with other potential antioxidants, such as ascorbic acid and trolox. In addition, it showed high healing capacity in wounds, biocompatibility and non-toxic properties [65].

Another application of lignin is as protection against cartilage degradation. Oxidative stress is one of the main causes of cartilage degradation leading to osteoarthritis. Nanofibrous membranes of polycaprolactone (PCL) and lignin, produced by electrospinning, attenuated oxidative stress and increased cell viability, reducing apoptosis and inflammation factors. These properties have been associated with protection against osteoarthritis as well as cartilage degradation [97].

Natural biopolymers such as chitosan and hyaluronic acid have been explored for providing an extra-cellular matrix that mimics a microenvironment with excellent cellular affinity. The combination of these matrices with a terpolymer containing catechol resulted in an interpenetrated polymeric network that prevents binding with cytotoxic agents. In addition, it allowed the controlled release of catechol into the medium. These hydrogels showed good adhesion, protection against oxidative stress and suspension of the anti-inflammatory process, biocompatibility, easy vascularization and potential for tissue regeneration and treatment of chronic wounds [98].

Antioxidant biomaterials have shown great application potential in grafts for synthetic neural tissues, instead of autologous grafts. In a study carried out by Puertas-Bartolomé et al. [99], PCL/lignin nanofibers showed good mechanical properties, high antioxidant capacity and biocompatibility. These factors promoted Schwann cell myelin and stimulated neurite outgrowth of DRG neurons, highlighting good potential for other tissue applications.

Table 3 presents antioxidant additives used in biomaterials and the various applications in biomedical devices found in the literature.

## 5. Conclusions

In the last decades, the growing interest in intercorrelating “bio” and “material” has brought innovations and opportunities for researchers. This review contributes with an overview of the common materials used for biomaterial development, the explored sources of bioactive compounds with antioxidant properties, the biological properties and the main applications of biomaterials reported in the literature.

Gelatin, cellulose, starches, chitin, chitosan, alginate and carrageenan are among the most explored materials used for biomaterial development, and they are able contribute to important material properties such as gelling, encapsulation, delivery of compounds, texture modification and film formation. Due to the role of oxidative stress as a key player in many disease conditions, marine organisms, microorganisms, plant extracts and other sources have been explored for the development of biomaterials with antioxidants, which may represent a relevant option for several areas, including materials science and the pharmaceutical, medical and food science fields. These materials have shown interesting properties, such as antimicrobial activity, anticancer effects, immunostimulant activity, anti-coagulation properties, inhibition of oxidative stress and inflammatory reactions and positive effects in wound healing. Another area that has been gaining attention is the application of antioxidant biomaterials as active, intelligent and biodegradable packaging films, overcoming a problem related to the utilization of materials from non-renewable sources.

Nevertheless, there is still a huge opportunity for researchers to find different materials that provide compounds with antioxidant properties, especially considering the widely available sources from food and agricultural wastes and byproducts. Further works could also explore blends between different materials to improve mechanical properties, texture quality, and the chemical and physical stability of the biomaterials. However, providing evidence for the biodegradability and biocompatibility of these biomaterials may be challenging, and it highlights the need for further works.

## Figures and Tables

**Figure 1 antioxidants-11-01644-f001:**
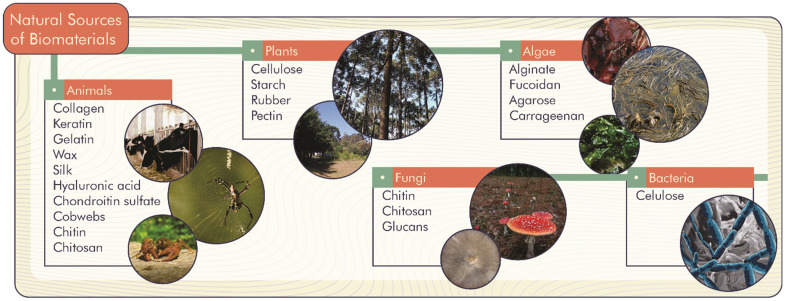
Primary natural sources of biomaterials.

**Figure 2 antioxidants-11-01644-f002:**
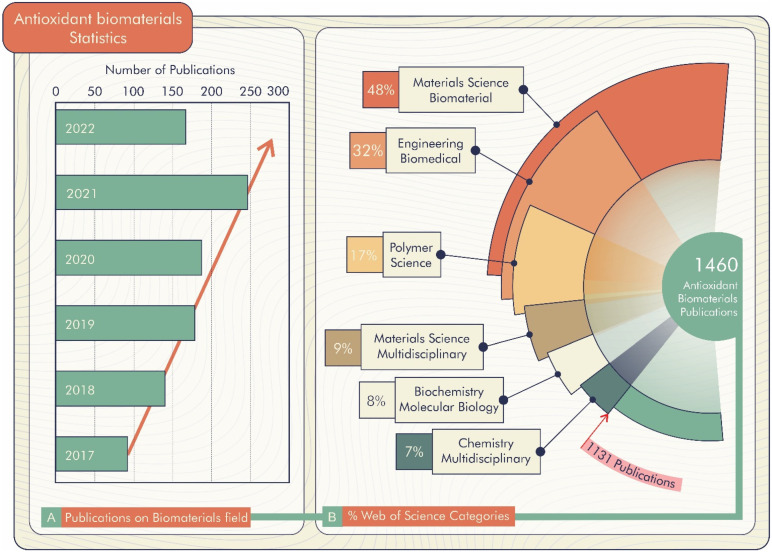
(**A**) Percentage of publications about biomaterials in research fields of Web of Science. (**B**) Publications on biomaterials field (18 April 2022).

**Figure 3 antioxidants-11-01644-f003:**
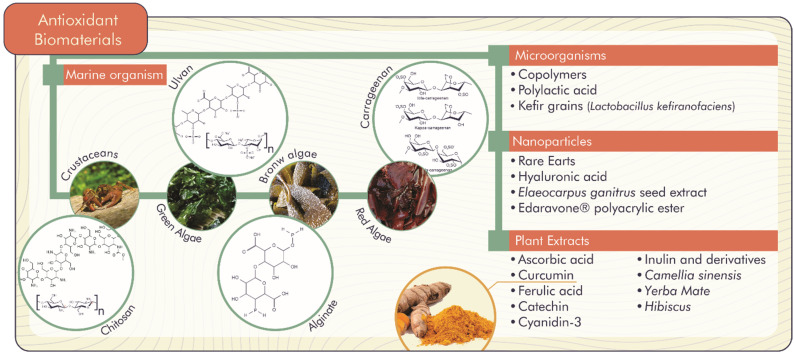
Natural antioxidant biomaterials with important biological functions.

**Figure 4 antioxidants-11-01644-f004:**
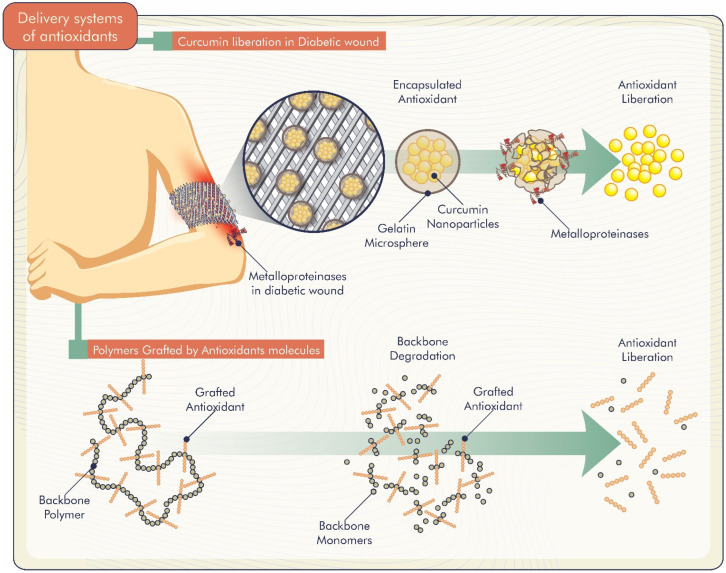
Controlled release mechanisms of antioxidants incorporated into biomaterials.

**Figure 5 antioxidants-11-01644-f005:**
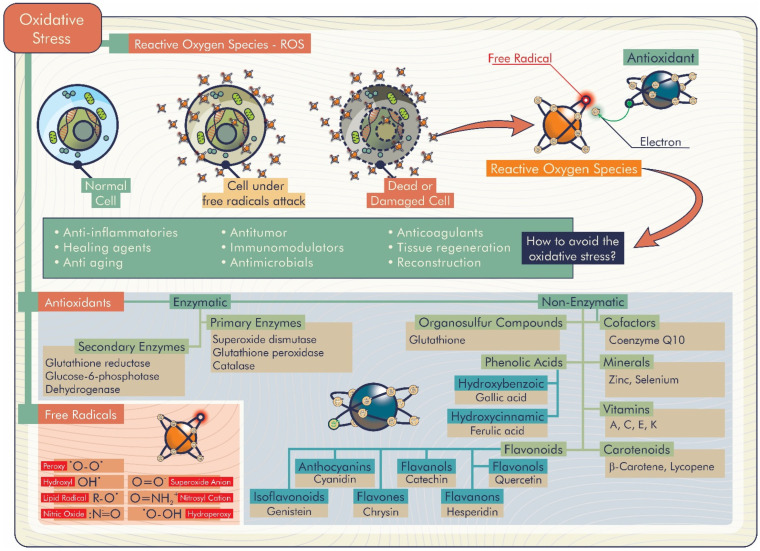
Important biological properties attributed to antioxidant biomaterials.

**Table 1 antioxidants-11-01644-t001:** Biological activities of antioxidant biomaterials.

Source of Biomaterial	Type of Biomaterial	Function/Effect	References
Alginate-chitosan and tetracycline gelatin	Hydrogel	Wound healing and antimicrobial activity against *Escherichia coli* and *Staphylococcus aureus*	Chen et al. [60]
Peptide conjugated with caffeic acid	Biocosmetic	Slows down the natural aging process	Lee et al. [61]
Curcumin	Hydrogel	Wound healing in diabetics	Liu et al. [58]
Jujube extract	Hydrogel	Wound healing in diabetics	Huang et al. [62]
Polyvinyl alcohol modified with citric acid (PVA-C)	Films	Inhibition of oxidative stress and lipopolysaccharide-induced inflammatory reactions	Wu et al. [59]
Alginate with Edaravone^®^	Hydrogel nanocomposite	Antioxidant effect and wound healing in diabetic mice	Fan et al. [51]
Chitosan	Film	Antimicrobial effect against *Staphylococcus aureus* and *Escherichia coli*	De Masi et al. [63]
Tannic acid and benzalkonium chloride	Catheters	Bactericidal activity against *Staphylococcus aureus* and *Escherichia coli*	Liu et al. [64]
Curcumin-encased hydroxyapatite nanoparticles	Encapsulated	Antimicrobial, antioxidant, and anticancer effects	Sebastianmmal et al. [48]
PEG, PPG e PDMS with lignin extracted from coconut husk	Nanogel	Accelerated healing of burns, reduced active oxygen level, and protected human hepatocyte cells against apoptosis	Xu et al. [65]
Modified dopamine	Hydrogel	Wound healing and skin burns in diabetics	Hu et al. [66]
Chitosan	Biopolymer	Anticarcinogenic and immunostimulating activity	Ivanova and Yaneva [22]
Chitosan	Hydrogel	Dentin-pulp regeneration of teeth and treatment of periodontitis	Fakhri et al. [67]
Tannic acid and collagen	Injectable spheres	Post-lumpectomy breast tissue reconstruction	Baldwin et al. [68]
Carrageenan	Film	Hydrophilicity and high ester sulfate content; anti-inflammatory, antitumor, antimicrobial, antioxidant, anti-hyperlipemic, anticoagulant and immunomodulatory properties	Wan et al. [28]
Ulvan	Hydrogel	Anti-coagulation, antioxidant, antibacterial, and anti-tumor properties	Wang et al. [27]
Chitosan with ulvan	Film	Antioxidant and whitening ability	Don et al. [36]
Chitooligosaccharide	Film	Antibacterial and antioxidant properties for wound healing	Jafari et al. [33]
*Elaeocarpus ganitrus* extract	Gold nanoparticles	Antibacterial, antioxidant and cytotoxic against a prostate cancer cell line	Vinay et al. [49]
Thrombospondin protein	Hydrogel	Resists oxidative stress damage, antioxidant, anti-photoaging, and wound healing effects	Wang et al. [29]
Chitosan with 3-Formylindole	Polymer	Intracellular ROS reducer, antioxidant, and antimicrobial properties	Ali et al. [32]
α-lipoic acid grafted with chitosan	Film	Antioxidant activity	Tan et al. [34]

**Table 2 antioxidants-11-01644-t002:** Antioxidant biomaterials from different polymer matrices and their applications in the food industry.

Antioxidant Additive		Source	Applications	References
Tannins	*Acacia*	Regenerated cellulose	Active packaging films	Huang et al. [72]
Phenolic compounds	Honey and pollen	*K*-carrageenan	Edible films for meat	Velásquez et al. [85]
Lignin	Rice husk	Poly(lactic acid)	Ultra-thin membranes for active packaging	Fontes et al. [75]
	Hop	Chitosan-silica	Functional films with active ingredient release for soybean oil storage packaging	Tian et al. [80]
	*Ceratonia siliqua* L.	Cellulose	Multi-layer packaging	Ait Ouahioune et al. [74]
	Pine nut shell	Maize starch	Aerogels for water absorption in packaging	Fonseca et al. [77]
	Moringa	Chitosan-guar gum-polyvinyl alcohol blends	Active films for packaging	Bhat et al. [82]
	Green tea	Polycaprolactone/poly(lactic acid)	Biodegradable active films for packaging	Sadeghi, Razavi, and Shaharampour [86]
	Pecan nut shell	Whey protein	Active films for packaging	Arciello et al. [87]
	Durian	Gelatin	Active films for packaging	Joanne Kam et al. [88]
Essential oils	Zataria	Ethyl cellulose/polycaprolactone/gelatin	Nanofibers for food packaging	Beikzadeh et al. [89]
	Lemon grass	Chitosan and starch	Biodegradable active film for packaging	Istiqomah et al. [90]
Others	Chitosan	Oxidized cellulose nanofiber	Active films for packaging	Soni et al. [79]
	Graphene oxide	Chitosan	Food packaging and biological applications	Barra et al. [78]
	Curcumin	Chitosan	Edible coating for pork	Shen et al. [84]
	Benzyl-isocyanate	Chitosan-cellulose nanocomposite	Active films for packaging	Jiang et al. [91]
	Thymol and/or carvacrol	Poly(lactic acid)/poly(ε-caprolactone)	Biodegradable films for active food packaging	Lukic, Vulic, and Ivanovic [92]
	Lysozyme	Pullulan	Functional food packaging films	Silva et al. [93]

**Table 3 antioxidants-11-01644-t003:** Antioxidant biomaterials from different polymer matrices and their applications in the food industry.

Antioxidant Additive		Matrix	Applications	References
Lignin	Lignin	Polyurethane copolymers of polyethylene glycol/polypropylene glycol/polydimethylsiloxane	Nanogel for wound healing	Xu et al. [94]
	Lignin	Poly(ε-caprolactone) nanofibers	Membrane for effective treatment of osteoarthritis	Liang et al. [97]
Plant extracts	Leaves of *Cinnamomum osmophloeum Kanehira*	-	Suppresses melanogenesis and protects against DNA damage	Ho, Wu, and Chang [95]
	Cassia alata extract nanocomposite/silver nanoparticle/montmorillonite	Cellulose nanofiber	Scaffold for wound regeneration	Subha et al. [100]
	Zinc oxide complex and grapefruit seed extract	Cellulose	Nanocomposite hydrogel films for potential applications in the treatment of chronic wounds	Dharmalingam and Anandalakshmi [101]
	*Tiliaplatyphyllos*	Chitosan	Scaffolds in tissue engineering	Radwan-Pragłowska et al. [102]
	Cannabidiol	Alginate/zinc	Multifunctional dressing to promote wound healing	Zheng et al. [103]
Others	Citric acid	Polyvinyl alcohol	Modification of polymers to biomaterial	Wu et al. [59]
	Derivatives of hydroxycinnamic acid (*p*-coumaric acid and ferulic acid)	Polyvinylpyrrolidone	Hydrogels as possible wound dressings	Contardi et al. [104]
	Silver nanoparticles	Chitosan/polyvinyl alcohol	Nanoparticles for skin healing dressings	Hajji et al. [105]
	Curcumin	Polyurethane	Hydrogel for potential use as dressings or tumor isolation membranes	Feng et al. [106]
	Polyorganophophazene	-	Microspheres for bone regeneration	Huang et al. [107]
	Hydroxyapatite/curcumin nanoparticles	-	Biomedical applications	Sebastiammal et al. [48]
	Hyaluronic acid/tannic acid	-	Hydrogel sunscreen	Gwak, Hong, and Park [98]

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
