# Peer review of "The Importance of Antioxidant Biomaterials in Human Health and Technological Innovation: A Review"

_antioxidants, 2022, doi:10.3390/antiox11091644_

Round 1
Reviewer 1 Report (Previous Reviewer 1)
It seems more acceptable now.
Author Response
Dear reviewer,
1. It seems more acceptable now. Reply: Thank you.Yours sincerely,
Alessandra Cristina Pedro, PhD
Embrapa Florestas
Non-wood products laboratory
Curitiba, PR, Brazil
Reviewer 2 Report (Previous Reviewer 2)
The authors should correctly categorize the materials. For example, the authors created “2.1.1. Antioxidant-carrying materials” in the revised manuscript. However, in actuality, hydrogels that formed recombinant thrombospondin antioxidant proteins (in lanes 175-178) may exhibit antioxidant properties themselves. I think this protein is not a carrier material. The category is wrong.
In addition, the authors should check references. For example, ref [27], which described a review paper about DDS carriers but not recombinant thrombospondin antioxidant protein. Also, corresponding the reference number in the text and reference list should be meticulously checked.
Author Response
Dear Reviewer,
All suggestions were accepted, more information and details were included in the text, and the manuscript was revised thoroughly. All the modifications performed in the revised manuscript are highlighted in red.
- The authors should correctly categorize the materials. For example, the authors created “2.1.1. Antioxidant-carrying materials” in the revised manuscript. However, in actuality, hydrogels that formed recombinant thrombospondin antioxidant proteins (in lanes 175-178) may exhibit antioxidant properties themselves. I think this protein is not a carrier material. The category is wrong.
Reply: Thank you for your consideration. The manuscript was revised and we modified this category.
- In addition, the authors should check references. For example, ref [27], which described a review paper about DDS carriers but not recombinant thrombospondin antioxidant protein. Also, corresponding the reference number in the text and reference list should be meticulously checked.
Reply: The references and their numbering have been checked and modified as requested.
Yours sincerely,
Alessandra Cristina Pedro, PhD
Embrapa Florestas
Non-wood products laboratory
Curitiba, PR, Brazil
Reviewer 3 Report (Previous Reviewer 3)
I suggest publication.
Author Response
Dear reviewer,
1. I suggest publication. Reply: Thank you.Yours sincerely,
Alessandra Cristina Pedro, PhD
Embrapa Florestas
Non-wood products laboratory
Curitiba, PR, Brazil
Round 2
Reviewer 2 Report (Previous Reviewer 2)
This paper is an important contribution and I recommend that it be accepted for publication.
This manuscript is a resubmission of an earlier submission. The following is a list of the peer review reports and author responses from that submission.
Round 1
Reviewer 1 Report
The manuscript “The importance of novel antioxidant biomaterials: a review of current evidence and future perspectives” by Pedro et al. reviews the main available biomaterials, antioxidant sources and assigned biological activities. Although authors described the importance of biomaterials and antioxidant biomaterials, these biomaterials were employed for a long-term and lack the novelty. Therefore, I would suggest rejection and resubmission possible of this work. Here are the comments and suggestions:
1. Authors are suggested to focus on the antioxidant materials.
2. The mechanisms of antioxidation of these biomaterials should be illustrated.
3. The novel antioxidant biomaterials should be added.
4. What is the point of Fig. 5? What would be the antioxidation effects of these materials?
5. Authors are suggested to give a more in-depth review and discussion on antioxidant biomaterials.
Reviewer 2 Report
The present manuscript seemed confusing because the definition of "antioxidant biomaterials" is unclear in the manuscript. For instance, the authors definite the biomaterials to "from natural sources such as animals, plants, fungi, algae and bacteria, being composed mainly of protein, lipid and carbohydrate molecules," but in actuality, biomaterials include artificial materials such as metal alloy, ceramics, polymers, and their composites. In addition, the updated definition of biomaterials also includes drugs and biological products, as the authors cited. Therefore, the manuscript made the impression that the review for antioxidant molecule DDS carriers derived from natural substances.
The authors should rearrange whole structure of manuscript.
The title is “The importance of novel antioxidant biomaterials,” however, it, unfortunately, is not found where is the importance.
The position of Figure 2 in this review is unclear. Biomaterials include various devices and substances for medical use. Therefore, the authors should show statistics of publications in natural biomaterials rather than the whole of biomaterials if the authors' interest is "biomaterials derived from the natural substance." If the authors' interest is "antioxidant substances," the authors should show statistics on antioxidants.
Marine organism and microorganisms extracts, listed as antioxidant biomaterials by the authors, are carrier materials for functionalizing antioxidant molecules. However, nanoparticles, a form or shape of materials, are described in 2.1.3. I recommended the authors rearrange these categories, especially in this section.
Reviewer 3 Report
Alessandra Cristina Pedro et al reviewed the antioxidant biomaterials and their sources, biological activities, and relative applications. I briefly went through the manuscript and feel that this manuscript deserves publication. A minor suggestion is that there are too many useless words throughout the manuscript. For example, Page 4 lines 150: ” Biomaterials based on antioxidant compounds have attracted the attention of researchers and have shown great potential for expansion in the pharmaceutical, biomedical, cosmetic and food areas.” There are also many “popular science language”. For example, Page 4 lines 150: “ In particular, because the accumulation of ROS, responsible for the lipid peroxidation of the hydroxyl radical, hydrogen and superoxide anion, are associated with changes and damage to cellular constituents (membrane lipids, nucleic acids and proteins), which signal several complications to human health”. For a scientific review, the reader may expect to learn the recent advances in the field rather than the popular science.